# An Efficient and Lightweight Convolutional Neural Network for Remote Sensing Image Scene Classification

**DOI:** 10.3390/s20071999

**Published:** 2020-04-02

**Authors:** Donghang Yu, Qing Xu, Haitao Guo, Chuan Zhao, Yuzhun Lin, Daoji Li

**Affiliations:** Institute of Geospatial Information, PLA Strategic Support Force Information Engineering University, Zheng Zhou 450001, China; dong_hang@aliyun.com (D.Y.); ghtgjp2002@163.com (H.G.); hehe549124@outlook.com (C.Z.); lyz120218@163.com (Y.L.); wang111@alumni.sjtu.edu.cn (D.L.)

**Keywords:** scene classification, remote sensing image, bilinear model, MobileNet, convolutional neural network

## Abstract

Classifying remote sensing images is vital for interpreting image content. Presently, remote sensing image scene classification methods using convolutional neural networks have drawbacks, including excessive parameters and heavy calculation costs. More efficient and lightweight CNNs have fewer parameters and calculations, but their classification performance is generally weaker. We propose a more efficient and lightweight convolutional neural network method to improve classification accuracy with a small training dataset. Inspired by fine-grained visual recognition, this study introduces a bilinear convolutional neural network model for scene classification. First, the lightweight convolutional neural network, MobileNetv2, is used to extract deep and abstract image features. Each feature is then transformed into two features with two different convolutional layers. The transformed features are subjected to Hadamard product operation to obtain an enhanced bilinear feature. Finally, the bilinear feature after pooling and normalization is used for classification. Experiments are performed on three widely used datasets: UC Merced, AID, and NWPU-RESISC45. Compared with other state-of-art methods, the proposed method has fewer parameters and calculations, while achieving higher accuracy. By including feature fusion with bilinear pooling, performance and accuracy for remote scene classification can greatly improve. This could be applied to any remote sensing image classification task.

## 1. Introduction

In recent years, with the development of Earth observation technology, remote sensing image resolution has continuously improved, datasets have become larger, and applications have continued to expand. Therefore, rapid and efficient interpretation of these images has important applications [1,2,3,4,5].

Classification and scene recognition are important methods for remote sensing image interpretation. Scene classification refers to dividing the image into blocks and labeling each with an appropriate category (such as residential areas, farmland, rivers, and forests) according to the makeup of the blocks. This is helpful for image management, retrieval, analysis, detection, and recognition of typical targets. When resolution increases, images become more diverse, allowing for fine-grained classification and identification. At the same time, the details of high-resolution remote sensing images are richer, the features in the image are more diverse, and the objects on the ground are usually staggered. The similarity between images of the same type decreases while the difference of same types increases significantly [6,7]. In addition, it is necessary to consider rotation and positional relationship among targets in the image. These problems bring challenges to high-precision scene classification.

At present, the classification methods with better performance can be broadly divided into two categories [3,4,8,9]: using handcrafted features and using deep learning.

1. Methods using handcrafted features. These methods mainly use handcrafted, low-level and middle-level features. Low-level features include color [10], gray-level co-occurrence matrix (GLCM) [11,12], local binary patterns (LBP) [13,14,15,16], texture features, Gabor descriptors [17], histogram of oriented graphic (HOG) features [18], and scale-invariant feature transform (SIFT) [19,20,21], etc. These features are easy to understand and implement, but cannot effectively represent high-level image semantic information. Using these features for image classification usually results in low accuracy. Therefore, low-level features are rarely used alone; they are usually used with middle-level features. The middle-level features are obtained by encoding the low-level features. This allows a relationship between the low-level features and the image semantic information to be established, improving classification performance. Bag of Visual Words (BOVW) is the most commonly adopted image classification method with middle-level features. Before the emergence of deep convolutional neural networks, BOVW had been the mainstream method for remote sensing image scene classification [22,23,24,25,26,27]. Spatial Pyramid Matching (SPM) [28], Spatial Co-occurrence Kernel (SCK) [29], Latent Dirichlet Allocation (LDA) [30,31], Probabilistic Latent Semantic Analysis (pLSA) [32], Fisher Kernel [33,34], Vector of Locally Aggregated Descriptors (VLAD) [35] and other technologies have also been widely used in remote sensing image scene classification for mining image semantic information. Compared with low-level features, scene classification methods using middle-level features have achieved higher accuracy. However, these methods require clever design features or specific constraints to increase feature discrimination. Many factors must be considered in feature design and their generalization is poor, especially in constructing middle-level features. The issues of information ambiguity and redundancy still lack effective solutions. The relationship between the middle-level features and semantic information has not been fully explored. It is hard for methods using handcrafted features to achieve high-precision image classification for complex scenes or massive data.

2. Methods using deep learning. Introducing deep learning, especially convolutional neural networks (CNN), into remote sensing image scene classification has greatly improved accuracy and efficiency. Until now, there have been two main methods of using CNN for scene classification: 1) using the CNN pretrained on ImageNet [36] for fine-tuning [3,37,38,39] or feature extraction; 2) improving the structure of existing CNNs [8,40,41,42,43,44,45,46], loss functions [46,47,48,49,50], or combining CNNs with handcrafted features [51,52,53,54,55]. Chen et al. [3] fine-tuned VGG16 [56] and achieved a far higher classification accuracy than methods using handcrafted features on large-scale remote sensing image datasets. Nogueira et al. [38] adopted three strategies (training from scratch, fine-tuning, and utilizing CNNs as feature extractors) on multiple CNNs to classify remote sensing image scenes. The results showed that all three strategies can achieve high accuracy, especially the one using CNN as a feature extractor before using a linear SVM classifier to retrain the extracted features. Chaib et al. [57] utilized the discriminative correlation analysis method to fuse features extracted by the CNN and the overall classification accuracy reached 92.96% when the training ratio is 20%. Cheng [47] and Goel [50], among others, used metric learning to improve feature distinguishability from different types of images. Zhu et al. [53] combined handcrafted features with high-level semantic features extracted by deep CNNs to improve the performance of scene classification, but this method cannot be trained and implemented in an end-to-end approach. Compared with traditional handcrafted features, training an existing deep CNN with remote sensing image data could achieve outstanding performance. Ameliorating the structure or loss function of CNNs also could improve classification accuracy. However, when considering a small training dataset (for example, when the training ratio is less than 20%), there are still many challenges for producing fast and high-precision classification for more diverse scenes.

Most current high-precision scene classification methods adopt deep CNNs (such as VGG16, GoogLeNet [58], ResNet50 [59], etc.) or add handcrafted features to the features extracted by deep CNNs. Generally, a deep CNN has numerous parameters requiring super-computing power to reduce calculation time. A deep CNN has disadvantages such as training difficulties, low efficiency, and poor applicability. Due to the addition of handcrafted features, the classification pipeline is also divided into two stages, which cannot be trained or implemented in an end-to-end manner. There are still few methods available for remote sensing image scene classification and object detection (such as methods proposed by Zhang et al. [60], Zhang et al. [61] Teimouri et al. [62], etc.,) so far.

In view of the above problems, therefore, this study introduces the idea of feature fusion in the bilinear model [63,64] with the CNN MobileNetv2 [65] and designs an efficient and lightweight CNN called BiMobileNet for remote sensing image classification. The architecture designed can not only achieve higher classification accuracy but also has fewer calculation numbers and parameters. The main contributions of this article are as follows:(1)The idea of a bilinear model in fine-grained visual recognition is introduced into remote sensing image classification, which enhances the ability of the CNN to identify different scene types. Compared with the state-of-the-art methods for remote sensing image scene classification, the proposed method can obtain superior performance.(2)By integrating the lightweight CNN MobileNetv2 and the feature fusion method of the bilinear model, the method in this study considers both the advantages of a lightweight structure and high accuracy. Compared with other state-of-the-art methods, the proposed architecture has fewer parameters and calculations. Therefore, image classification speed will be higher, rendering it more viable for production purposes and applications.(3)This study proposes that both the accuracy and complexity of the method should be considered simultaneously during classification. The method should be evaluated comprehensively in three aspects: accuracy, parameter, and calculation. In addition, we find that most methods use the UC Merced dataset with a training ratio of 80%, and the classification accuracy is close to saturation. We provide an accuracy benchmark when the training ratio is less than 30%.

The remainder of this study is organized as follows. In Section 2, we illustrate the datasets used and the proposed architecture in detail. In Section 3, results and analysis of experiments on several datasets are detailed. Section 4 discusses results and Section 5 concludes the study with a summary of our method.

## 2. Materials and Methods

### 2.1. Materials

In order to verify the effectiveness of BiMobileNet in remote sensing image scene classification, three datasets (UC Merced [4], AID [2], and NWPU-RESISC45 [3]) were used.

The images in the UC Merced dataset are selected from aerial images in the U.S. Geological Survey (USGS) national city map. The dataset contains 21 scene types, such as farmland, residential area, forest, and oil tank. Each scene type consists of 100 images with a size of 256 × 256 pixels. In total, this dataset consists of 2100 RGB images with spatial resolution of ~0.3 m. Figure 1 shows examples from this dataset. More information on the dataset can be found at http://vision.ucmerced.edu/datasets.

Compared with the UC Merced dataset, the AID dataset extends the number of scene categories to 30, with categories more finely classified. Each category contains ~220 to 440 RGB images; the total number of images in the dataset is 10,000. Image size is 600 × 600 pixels and the resolution is ~0.5–8 m. Figure 2 shows representative images of each class. More detailed information on this dataset can be found at http://www.lmars.whu.edu.cn/xia/AID-project.html.

NWPU-RESISC45 is a large-scale remote sensing image dataset, which further expands the number of categories and images. It contains 45 categories, each consisting of 700 RGB images with a size of 256 × 256 pixels. Image resolution ranges from 0.2–30 m. In addition, images cover more than 100 countries and regions, including different weather, seasons, spatial resolution, and occlusion factors. Compared with other datasets, NWPU-RESISC45 images are more complex and diverse. Figure 3 shows representative images of each class. More detailed information can be found at http://www.escience.cn/people/JunweiHan/NWPU-RESISC45.html.

Table 1 summarizes the three datasets. Given the differences between each dataset, experiments on each will help verify the robustness and generalization of our proposed method.

### 2.2. Method

The method in this study integrates the idea of a lightweight CNN MobileNetv2 and a bilinear model for fine-grained visual recognition. MobileNet [66] is a lightweight CNN proposed to apply deep learning on mobile and edge devices. It greatly reduces CNN parameters and calculations by using depthwise separable convolution. Although the classification accuracy of MobileNet on ImageNet is slightly lower than that of deep CNNs such as ResNet50, it has the unique advantages of a smaller size, fewer parameters, fewer calculations and can be used on mobile and embedded devices. MobileNetv2 introduced an inverted residual and linear bottleneck, further compressing parameters and calculations, improving performance. The bilinear model is a widely used method for fine-grained visual recognition. In the bilinear model, two parallel CNNs (which are separated by the last fully connected layers and classification layers) are used as feature extractors to obtain two deep features of the same image. The two features are then in a bilinear pooling instead of a connection, summation, or maximum pooling. Bilinear pooling is an efficient feature fusion strategy. Along with its concise form and gradient calculation method, the bilinear model can also be trained end-to-end and has excellent classification performance in fine-grained visual recognition. In the following sections, we introduce the depthwise separable convolution, linear bottleneck, inverse residual, bilinear model, and the network architecture.

#### 2.2.1. Depthwise Separable Convolution

The core idea of depthwise separable convolution (Figure 4) is to divide the traditional standard convolution operation into two steps: depthwise convolution and pointwise convolution. Assuming that the size of the input feature maps is *D_K_* × *D_K_* × *M*, using *N* convolution kernels of size *D_K_* × *D_K_* × *M* to perform the convolution operation on the input feature maps, *N* feature maps of size *D_R_* × *D_R_* can be directly obtained, where *D_R_* is the width and height of the input feature maps, *M* is the number of channels of the input feature maps, *D_K_* is the width and height of the convolution kernels, *N* is the number of convolution kernels, and *D_R_* is the width and height of the output feature maps. When using depthwise separable convolution to operate a convolution on feature maps of size *D_R_* × *D_R_* × *M*, firstly *M* convolution kernels of size *D_K_* × *D_K_* × 1 are used to convolve with each channel of the feature maps separately. The size of the output feature maps is *D_R_* × *D_R_* × *M*. Depthwise convolution only changes the width and height of the original feature maps but does not change the number of channels. To increase the channels of the feature maps, pointwise convolution can be used after depthwise convolution. In the process of pointwise convolution, *N* convolution kernels of size 1 × 1 × *M* are used to operate convolution on the feature maps to obtain *N* feature maps of size *D_R_* × *D_R_*. Finally, the size of the feature maps generated by standard convolution and by depthwise separable convolution are the same, but the number of parameters and calculations has changed. The standard convolution calculation is *D_K_* × *D_K_* × *M* × *N* × *D_F_* × *D_F,_* while the depthwise separable convolution calculation is *D_K_* × *D_K_* × *M* × *D_F_* × *D_F_* + *M* × *N* × *D_F_* × *D_F_*. In a CNN, when the size of the convolution kernels is 3 × 3, the depthwise separable convolution calculation is ~1/9 of the standard convolution calculation. In addition, in order to further reduce the network parameters, MobileNet introduces two parameters: channel multiplier and resolution multiplier. The channel multiplier, α, is used to proportionally expand or reduce the number of feature channels; the resolution multiplier, *ρ,* is used to proportionally enlarge or reduce the size of the feature maps. The calculation of the depthwise separable convolution after reducing the channel number and size of feature maps is *D_K_* × *D_K_* × *αM* × *ρD_F_* × *ρD_F_* + *αM* × *αN* × *ρD_F_* × *ρD_F_*.

#### 2.2.2. Linear Bottleneck

The introduction of the linear bottleneck is to solve the information loss caused by using activation functions such as ReLu (rectified linear unit) in CNNs. The activation function in a CNN generally performs a non-linear transformation on the feature maps of the input. The non-linearity allows the neural network to approximate any arbitrary non-linear function and enhances the network’s ability to express information. The ReLu activation function outputs are zero if the input is negative; for positive inputs, the output is dependent on a linear transformation. The ReLu function, therefore, increases the sparsity of the network (outputs of zero are ignored) and reduces the interdependence between parameters, thus reducing the possibility of model overfitting. However, the ReLu function will also cause large losses of information for features with small channels, as the process of feature dimension reduction is also a process of feature compression. The essence of the linear bottleneck, therefore, is that after the pointwise convolutional layer and the batch normalization layer, the feature maps are directly passed to the next convolutional layer without using a non-linear activation function (Figure 5).

#### 2.2.3. Inverted Residual Block

MobileNetv2 uses the feature shortcut connection idea in the ResNet structure to fuse feature maps between different convolutional layers (Figure 6). When ResNet performs shortcut feature connection, it first uses pointwise convolution to compress the channel number of the input feature maps (usually to 0.25 times the original number). Feature maps after compression are passed to a standard convolution module where the channel number of the feature maps is in constant. The number of channels is restored to the original number using another pointwise convolution. Finally, the feature maps are added to the input feature maps. The inverted residual adopted in MobileNetv2 is the opposite: a complete inverted residual structure first performs a pointwise convolution to expand the number of feature channels to *m* times the original number (*m* is an integer greater than 1; in MobileNetv2 the value of *m* is 6), and then performs depthwise and pointwise convolution. In the second pointwise convolution, the feature map channels are expanded to the original number and then the obtained feature maps are added to the original feature maps. Similarly, the ReLu activation function is no longer used after the second pointwise convolution. The design of the inverted residual structure not only has good memory efficiency, but also improves network performance.

#### 2.2.4. Bilinear Model

Lin et al. [63] first proposed a bilinear CNN model (B-CNN) in fine-grained visual recognition tasks producing excellent performance. The core idea is to use two parallel CNNs to extract features from the same image and then merge the two features using bilinear pooling to obtain a new feature vector (Figure 7). A standard bilinear model **ℬ** consists of four components: **ℬ** = (*f*_A_, *f*_B_, 𝒫, 𝒞), where *f*_A_ and *f*_B_ are two feature extraction functions based on CNNs, which are used to extract features of the same image, 𝒫 is a pooling function, and 𝒞 is a classification function. When two CNNs extract features from the same image ***I*** and perform bilinear pooling in position *l*, the outer product operation is used, and the calculation process is as follows:(1)$$b\left(l,I,f_{A},f_{B}\right)=f_{A}^{T}\left(l,I\right)f_{B}^{T}\left(l,I\right),$$
(2)$$\xi \left(I\right)=\underset{l}{\sum }b\left(l,I,f_{A},f_{B}\right),$$
(3)x=vec(ξ(I)),
(4)$$y=\mathrm{sign}\left(x\right)\sqrt{\left|x\right|},$$
(5)$$z=y/{\left|\left|y\right|\right|}_{2}.$$
when the size of output feature maps of the input image was *D_W_* × *D_H_* × *C*, feature maps of size *D_W_* × *D_H_* × *C*^2^ could be obtained through an outer product operation. Each feature map was then summed and pooled globally to obtain a bilinear feature ***x*** of size *C*^2^; a square root operation (Equation (4)) and a normalization operation (Equation (5)) on the bilinear feature were then carried out to obtain a bilinear vector. Finally, logistic regression or a support vector machine (SVM) was used for classification with the bilinear vector. The bilinear model is not only simple in form and procedure but also enables end-to-end training and testing. It also has excellent performance in fine-grained classification tasks.

### 2.3. Proposed Architecture

The integration of the structure of MobileNetv2 and the method of feature fusion in the bilinear model, the network structure of BiMobileNet, is shown in Figure 8. The network includes main three parts: feature extraction, bilinear pooling of features to obtain a bilinear vector, and classification of bilinear features.

The backbone network of the feature extraction layer utilizes MobileNetv2 but does not use all layers. Instead, we removed the last three layers: one convolutional layer, one average pooling layer, and one classification layer. This is because the size of feature maps of the last convolutional layer is 7 × 7 × 1280. This still contained too many parameters; we wanted as few as possible. The structure information of BiMobileNetv2 is shown in Table 2. The feature extraction layer contains a convolutional layer and seven bottlenecks. Every bottleneck consisted of a linear or inverted residual block. For example, when an image with size 224 × 224 × 3 passes through the feature extraction layer, feature maps with size of 7 × 7 × 320 are obtained. In BiMobileNet we did not use two convolutional neural networks to extract features. Instead the feature maps extracted from the same network were shared to reduce the parameters and calculations in the model.

Before bilinear fusion of the obtained features, inspired by the hierarchical bilinear model proposed in [64], we used two feature transformation layers on the extracted features, thereby transforming the same feature into two different, but similar, features. The feature transformation layer is essentially a convolutional layer, with a size of 1024 convolution kernels of size *k* × *k* (here *k* is 3). After the feature maps with size 7 × 7 × 320 pass through the feature transformation layer, two kinds of feature maps with size 7 × 7 × 1024 are generated. The original bilinear model used the outer product operation when performing bilinear pooling operations on two different features. This expanded the dimensions of the obtained feature maps by *C* times (*C* is the channel number of feature maps); the dimensions of the bilinear feature vector were also expanded *C* times. When the channel number of the two features is 1024, million-dimensional feature vectors were generated. Not only is the model prone to overfitting, but also the training and implementation time is significant. Using the Hadamard product, instead of the outer product, in the bilinear model keeps the dimensions of the original feature maps unchanged. The Hadamard product is the multiplication of the elements at the corresponding positions in the two matrices of the same order and does not change the dimension of the matrices. In the bilinear fusion in BiMobileNet, two feature maps with size 7 × 7 × 1024 are processed by the Hadamard product to generate bilinear features with size 7 × 7 × 1024, keeping the feature dimensions unchanged. The average pooling operation is performed on the bilinear features with the size 7 × 7 × 1024; a bilinear feature vector with a size of 1 × 1 × 1024 is obtained. The classification layer is a fully connected layer. Its input is a bilinear feature vector with size of 1 × 1 × 1024 and the probability of each category is the output.

### 2.4. Experimental Setup

#### 2.4.1. Implementation Details

Before the experiment, each dataset was divided into training and test sets. In order for comparisons with other methods, we adopted different training ratios for the experiments. The training and test sets were chosen randomly from the original dataset. Every experiment was performed five times. The mean and standard deviation of the five results were calculated. By rotating the training images 90°, 180°, and 270° clockwise, horizontal flip and vertical flip, we expanded the training data six-fold. This augmentation helped to generalize the CNN models.

We utilized the open source deep learning framework PyTorch to build BiMobileNet. BiMobileNet can be trained end-to-end, using stochastic gradient descent to update its parameters. As BiMobileNet shares part of the MobileNetv2 network structure, before training, we used MobileNetv2’s pre-trained weight on ImageNet to initialize parameters. The hyperparameter settings in BiMobileNet were as follows: the initial learning rate of the feature extraction layers was 0.01, while the initial learning rate of the bilinear pooling layer and the classification layer was 0.1. Every 10 epochs, the learning rate was reduced by 0.5 times. The momentum and weight decay were 0.9 and 0.0005, respectively. The training batch size was 32, and the number of training epochs was 100. We use the well-trained model whose training loss is stable to predict the test set. All experiments were performed on a device with Intel Core i7-6900K CPU 64-GB RAM and GeForce GTX1080Ti GPU 11-GB RAM.

#### 2.4.2. Evaluation Protocol

BiMobileNet was comprehensively evaluated from aspects of classification accuracy and model complexity. The accuracy was expressed by two criteria: confusion matrix and overall accuracy. In the confusion matrix, each row represents the true class and each column represents its predicted category. The elements on the diagonal in the confusion matrix represent the classification accuracy of one class and the elements on the non-diagonal CM*_ij_* represent the probability that the images from class *i*th are mistakenly recognized as class *j*th. The overall accuracy is defined as the number of correctly predicted images divided by the total number of predicted images.

The model complexity includes time complexity and space complexity. Time complexity indicates the number of model operations, determines the training and prediction time of the model, and is represented by floating point operations (FLOPs). The higher the time complexity, the slower the model speed. For the model with high time complexity, model training and prediction time is long, which is less favorable for practical training and application. Space complexity indicates the space size of the model, which can be represented by the model size and the total number of model parameters. A model with considerable parameters needs a large amount of data to train and it is very easy to over fit.

## 3. Results

### 3.1. Classification of the UC Merced Dataset

The training ratios were set at 20%, 50%, and 80%. Table 3 shows the classification performance comparison of our architecture compared to the state-of-the-art methods on the UC Merced dataset. By analyzing the overall accuracies obtained by state-of-the-art methods under different training ratios, we find that on the UC Merced dataset, when using 80% of the data for training, the overall accuracy of other state-of-the-art methods are very close to 99.00%. When the training ratio is 80%, the classification accuracy becomes saturated, and it is difficult to improve it further. Using 80% of data for training, BiMobileNet nearly achieves the highest overall accuracy (99.03% compared to 99.05%); using 50% of data for training, BiMobileNet achieves the highest overall accuracy of 98.45% — this is better than many methods which use a training ratio of 80%. When training with 20% data, the classification accuracy of BiMobileNet reaches 96.41%. On one hand, BiMobileNet achieves outstanding performance when the training ratio is 50% (similar result to 80% training ratio); however, it also demonstrates that the classification accuracy is saturated when the training ratio is 80%, and it is difficult to further improve accuracy. We must consider model complexity and other issues with a small training sample.

Most of the state-of-the-art methods in Table 3 adopt deep CNNs (such as VGG16, ResNet50, etc.). Generally, deep CNNs with many layers and parameters have a large calculation component (the parameters and calculation are discussed in detail in Section 3.3), and the training of these networks requires a large amount of data. Therefore, it is difficult to train these networks with a small amount of training data, and overfitting often occurs. In addition, for a classification task, it is unreasonable and unrealistic to use 80% of the data for training. This is because annotation is a time-consuming job, and manually labeling 80% of the data is unfeasible. For practical applications, it is necessary to reduce manual annotation as much as possible and improve classification accuracy and efficiency with as little data as possible. However, BiMobileNet can still achieve an accuracy of 96.41% when only 20% of the data is used for training. In order to further verify BiMobileNet performance with little training data, more training ratios (5%, 10%, 15%, 20% and 25%) were used (Table 4). When the training ratio was 5% (e.g., five images randomly selected from each category for training and 95 images are predicted), BiMobileNet achieved an amazing overall accuracy of 86.74%, which was far higher than the results of fine-tuning VGG16, ResNet50 and MobileNetv2 directly. That means when faced with a new larger dataset, we can only label a very small portion of the data for training and then predict the remaining data, which can save a lot of labor and time. When the training ratio was 20%, BiMobileNet achieved an overall accuracy of 96.41% while the method proposed by Chaib [57] reached 92.96%. For the other lower training ratios, BiMobileNet also achieved excellent performance. This demonstrates that the method in this study is not prone to overfitting when using little training data, and has a large accuracy advantage over the deep CNNs in scene classification task using the UC Merced dataset.

Figure 9, Figure 10, Figure 11 and Figure 12 show the confusion matrices for training ratios of 5%, 10%, 20%, and 50%, respectively, on the UC Merced dataset. When the training ratio is 5%, 11 of the 21 scene categories achieve a classification accuracy greater than 92%, and only four categories: buildings (0.73), dense residential (0.33), intersection (0.76), and river (0.57), are lower than 80%. Many dense residential images (which have the lowest accuracy) are misidentified as medium residential and mobile home park, as the three types are very similar. With little training data (5 images), it is difficult to distinguish the three categories effectively for a CNN. When the training ratio is 10%, 16 of the 21 scene categories achieve a classification accuracy of greater than 92%; dense residential has the lowest accuracy (67%) but twice as high compared to the 5% training ratio. When the training ratio is 20%, 16 of the 21 scene categories achieve a classification accuracy of greater than 95%. When the training ratio is 50%, 18 of the 21 scene categories achieve a classification accuracy of greater than 98%; the three lower accuracy categories (building, dense residential, and medium residential) still achieve 92%. These three categories have poorer accuracy, because buildings are the main image component. Images were originally annotated based on building density, but this is a subjective and perceptual judgment, with no quantitative standard explaining the lower model accuracy.

### 3.2. Classification of the AID Dataset

When using the AID dataset, training ratios were set at 10%, 20%, and 50%. Table 5 compares the accuracy of state-of-the-art methods with our approach. Using a training ratio of 50%, the overall classification accuracy of BiMobileNet is 96.87%, which is higher than most other methods. When the training ratio is 20%, the overall accuracy is 94.83%, which is ~1% higher than all other methods. When the training ratio is 10%, the overall accuracy is 92.77%. BiMobileNet produces a similar accuracy to D-CNN [47], GCFs+LOFs [9] and SF-CNN [44] when using a training ratio of 50% but performs ~4.0%, ~2.5%, and ~1.2%, respectively, higher when the training ratio is 20%. The D-CNN, GCFs+LOFs, and SF-CNN networks all adopt VGG16, the parameters and calculations of which are much larger than MobileNetv2 utilized in BiMobileNet. 

Comparing Table 3 and Table 5, it is noted that the classification accuracy of different methods on the AID dataset is generally lower than the UC Merced dataset for a given training ratio. This is because the AID dataset has more categories, and the data is more diverse, rendering classification more challenging. Figure 13, Figure 14 and Figure 15 show the confusion matrices when the training ratios of the AID dataset are set to 10%, 20%, and 50%, respectively. When the training ratios are 10% and 20%, 22 and 27 categories (out of 30), respectively, have a classification accuracy greater than 91%. When the training ratio is 50%, the accuracy of most categories is greater than 98%. As training data increases, the classification accuracy of most categories improves significantly. However, the accuracy of the resort class is 76% (10% training ratio), 72% (20%), and 83% (50%)—lower than all other classes. Some images from the resort category are mistaken for park. This is mainly because park and resort have a similar object (buildings, vegetation) distribution. In addition, school and commercial, and center and square have similar features. Consequently, the school and resort classes have relatively low classification accuracies compared with other categories when the training ratio of the AID dataset is set to 50%. 

### 3.3. Classification of the NWPU-RESISC45 Dataset

For the NWPU-RESISC45 dataset, training data ratios were set to 10% and 20%. The classification accuracies of state-of-the-art methods and BiMobileNet are shown in Table 6. The NWPU-RESISC45 dataset has more categories and images than the other two datasets. The overall accuracy of BiMobileNet is 92.06% and 94.08% when the training ratios are 10% and 20%, respectively; this is higher than all but one other methods. When the training ratio is 10%, BiMobileNet accuracy is 2.1%,1.0% and 0.3% higher than SF-CNN [44], GLANet [46] and DML [49], respectively, and is similar to DDRL-AM [41]. SF-CNN, GLANet, and DML adopt deep CNN VGGNet; DDRL-AM adopts deep CNN ResNet18. The parameters and calculation of these two networks are significantly larger than MobieNetv2 used in BiMobileNet. In addition, when the training ratio is 20%, BiMobileNet accuracy is ~1.5%, ~0.6%, ~0.6%, and ~1.6% better than SF-CNN, GLANet, DML and DDRL-AM, respectively.

Figure 16 and Figure 17 show the confusion matrixes obtained by BiMobileNet using training ratios of 10% and 20%, respectively, on the NWPU-RESISC45 dataset. When the training ratio is 10%, the classification accuracy of 35 categories is greater than 90%. When the training ratio is 20%, the classification accuracy of 41 categories is greater than 90%; for GLANet, 38 categories are greater than 90%. As the training ratio increases, classification accuracy of most categories significantly improves. Although the accuracy of the church (72% with 10% ratio, 75% with 20% ratio) and palace (68%, 78%) categories improve, the accuracy is still significantly lower than other categories. This is because the two categories have similar architectural styles and layouts that can easily be misclassified.

## 4. Discussion

Currently, most remote sensing image scene classification methods only take classification accuracy into account, and seldom consider parameters, calculations and other issues. From our experimental results on three datasets (Table 3, Table 5 and Table 6), the majority of methods use deep CNNs such as VGG16 [56], GoogLeNet [58], and ResNet [59]. Although deep CNNs show strong generalization for image classification and object detection, they have clear disadvantages such as too many parameters and significant computation time for training and predictive processes. For example, the model size of VGG16 exceeds 512 MB, and the number of parameters exceeds 134 million. These directly affect the training and prediction time. More importantly, deep CNNs are easy to over fit with little sample data. Moreover, such deep CNNs can only be trained and implemented on hardware devices with high computational performance; this is not always conducive to practical application and deployment. Although CNNs such as GoogLeNet and ResNet have significantly reduced parameters and computational cost compared with VGG16, their parameters and computation are still significant and not suitable for mobile or other edge devices. In this case, the model with fewer parameters and less computation is more suitable for practical application, especially for real-time classification and object detection. In other words, the deployment of efficient and lightweight CNNs don’t not require high-end equipment and can also achieve better performance. For the task of remote sensing image classification, we may need to deploy the model on UAV (unmanned aerial vehicle), small satellite and other devices in the future to achieve real-time classification. Therefore, the model we designed not only needs to have outstanding performance, but also needs to focus on faster speed and less computation. That’s the advantage of our approach using MobileNet.

Comparison of different state-of-the-art methods on overall accuracy, parameters, calculation and model size are shown in Table 7. Many algorithms were improved on VGG16 or ResNet such as SF-CNN [44], DML [49], etc. As can be seen from the Table 7, VGG16 has the most parameters and calculations. SF-CNN replaced the last two fully connected layers with convolutional layers and adopted global mean pooling in the classification layer in VGG16. SF-CNN reduced the number of parameters but did not fundamentally reduce the calculations compared to VGG16. DML [49] did not change the structure of VGG16 but adopted mean center loss. Although it improved the accuracy of the original VGG16 in remote sensing image scene classification, it did not change the structure of the VGG16, and the number of parameters and calculations did not decrease. SAL-TS-Net [8] merged the features from two GoogLeNet networks in parallel. Compared with VGG16, the parameters and computational cost are reduced, but the precision is lower than directly fine-tuning VGG16. The BiMobileNet we designed has the least parameter and computation but achieved higher accuracies. Different channel reduction factors λ are set in BiMobileNet to further reduce model parameters and calculations. When λ is 0.75 and *k* is 1, the overall accuracy of BiMobileNet is higher than that of most methods. The number of parameters is approximately 1/11, 1/85, and 1/6 that of SF-CNN, DML and SAL-TS-Net, respectively. The calculation is approximately 1/65, 1/65 and 1/6 that of SF-CNN, DML, and SAL-TS-Net, respectively. Compared with other state-of-the-art methods, obviously BiMobileNet not only has outstanding performance, but also significantly reduces parameters and calculational cost.

## 5. Conclusions

This study introduces the idea of a bilinear model in fine-grained image classification into the remote sensing image scene classification task. Based on MobileNetv2, a highly efficient lightweight convolutional neural network (CNN) for remote sensing image scene classification is proposed – BiMobileNet. MobileNetv2 has the advantages of fewer parameters and a smaller number of calculations, but its remote sensing image classification performance is generally weaker than deep CNNs. MobileNetv2′s backbone network is used to extract the features of the images, with the features bilinearly pooled to increase intra-class consistency and inter-class distinction which can significantly improve the accuracy of scene classification and be applied to any remote sensing classification task. By training and testing on three widely used large-scale remote sensing image datasets, both the accuracy and complexity of the model were evaluated with the following conclusions drawn:1The accuracy of BiMobileNet in remote sensing image scene classification surpasses most state-of-the-art methods, particularly with little training data.2BiMobileNet requires fewer parameters and calculations making training and prediction faster and more efficient.3The challenges of remote sensing image scene classification are intra-class inconsistency and inter-class indistinction. The method of using bilinear pooling overcomes some of the difficulties of scene classification providing a simple and efficient method for scene classification.

In addition, compared with the ImageNet dataset (with 1000 categories), current remote sensing image datasets have far fewer categories (NWPU-RESISC45 dataset has 45 categories, the AID dataset has 30 categories, and the UC Merced dataset has 21 categories). Image categories are more diverse and complex than this, limiting practical applications. However, the use of the lightweight and efficient CNN described in this study will aid faster and more accurate classification of remote sensing images.

## Figures and Tables

**Figure 1 sensors-20-01999-f001:**
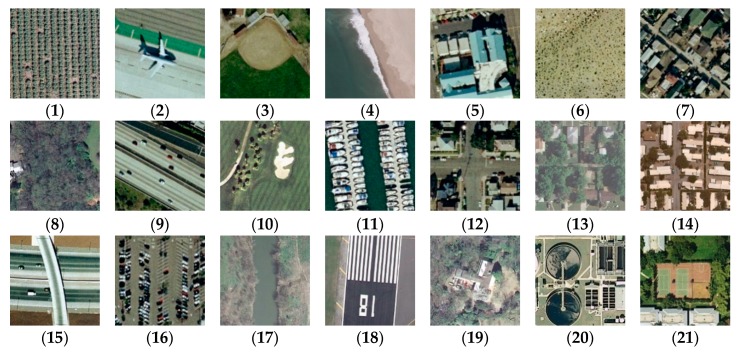
Class representatives of the UC Merced dataset: (**1**) agricultural; (**2**) airplane; (**3**) baseball diamond; (**4**) beach; (**5**) buildings; (**6**) chaparral; (**7**) dense residential; (**8**) forest; (**9**) freeway; (**10**) golf course; (**11**) harbor; (**12**) intersection; (**13**) medium residential; (**14**) mobile home park; (**15**) overpass; (**16**) parking lot; (**17**) river; (**18**) runway; (**19**) sparse residential; (**20**) storage tanks; and (**21**) tennis court.

**Figure 2 sensors-20-01999-f002:**
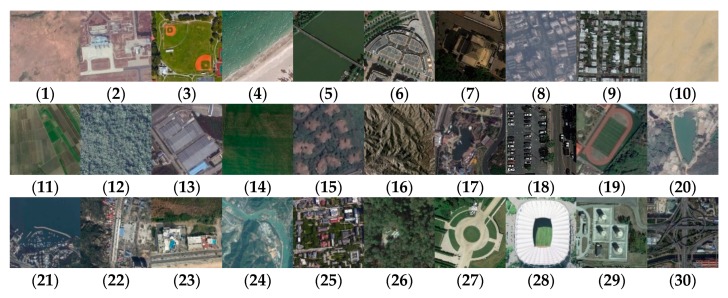
Class representatives of the AID dataset: (**1**) airport; (**2**) bareland; (**3**) baseball field; (**4**) beach; (**5**) bridge; (**6**) center; (**7**) church; (**8**) commercial; (**9**) dense residential; (**10**) desert; (**11**) farmland; (**12**) forest; (13) industrial; industrial; (**14**) meadow; (**15**) medium residential; (**16**) mountain; (**17**) park; (**18**) parking; (**19**) playground; (**20**) pond; (**21**) port; (**22**) railway station; (**23**) resort; (**24**) river; (**25**) school; (**26**) sparse residential; (**27**) square; (**28**) stadium; (**29**) storage tanks; (**30**) viaduct.

**Figure 3 sensors-20-01999-f003:**
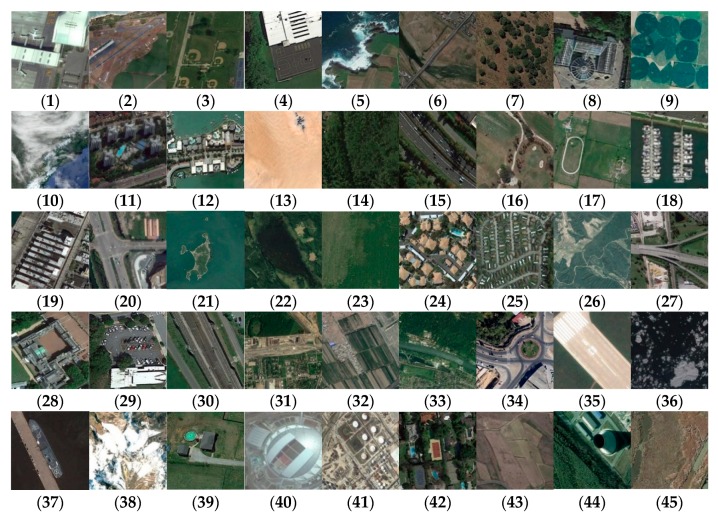
Class representatives of NWPU-RESISC45 dataset: (**1**) airplane; (**2**) airport; (**3**) baseball diamond; (**4**) basketball court; (**5**) beach; (**6**) bridge; (**7**) chaparral; (**8**) church; (**9**) circular farmland; (**10**) cloud; (**11**) commercial area; (**12**) dense residential; (**13**) desert; (**14**) forest; (**15**) freeway; (**16**) golf course; (**17**) ground track field; (**18**) harbor; (**19**) industrial area; (**20**) intersection; (**21**) island; (**22**) lake; (**23**) meadow; (**24**) medium residential; (**25**) mobile home park; (**26**) mountain; (**27**) overpass; (**28**) palace; (**29**) parking lot; (**30**) railway; (**31**) railway station; (**32**) rectangular farmland; (**33**) river; (**34**) roundabout; (**35**) runway; (**36**) sea ice; (**37**) ship; (**38**) snow berg; (**39**) sparse residential; (**40**) stadium; (**41**) storage tanks; (**42**) tennis court; (**43**) terrace; (**44**) thermal power station; (**45**) wetland.

**Figure 4 sensors-20-01999-f004:**
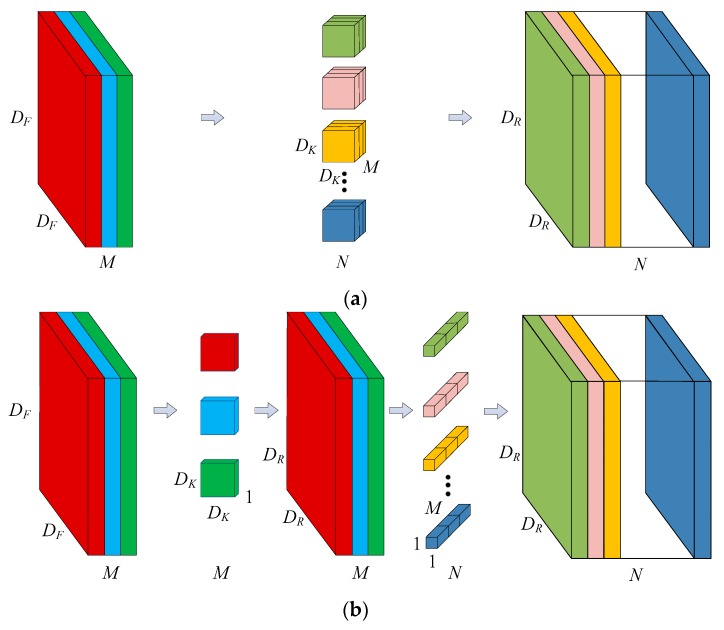
The difference between (**a**) standard convolution and (**b**) depthwise separable convolution. *D_K_* is the width and height of the convolution kernels, *D_R_* is the width and height of the output feature maps, *M* is the channel number of the input feature maps and *N* is the channel number of the output feature maps.

**Figure 5 sensors-20-01999-f005:**
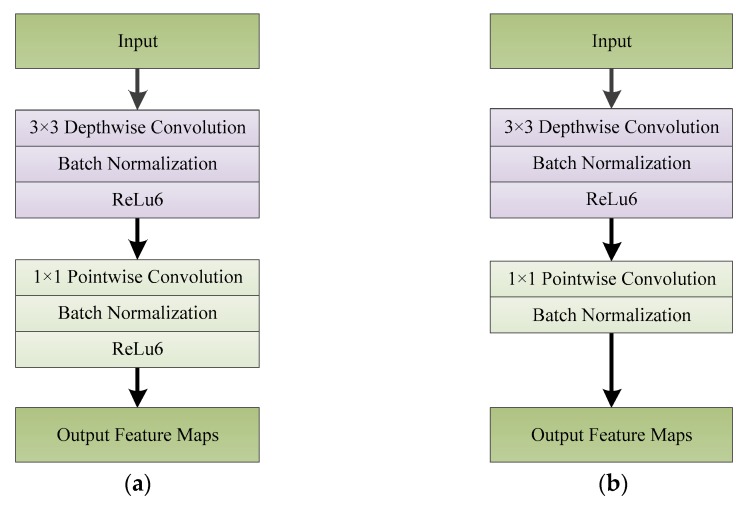
The basic structure of MobileNet: (**a**) unit in MobileNetv1 [66]; (**b**) unit in MobileNetv2 [65].

**Figure 6 sensors-20-01999-f006:**
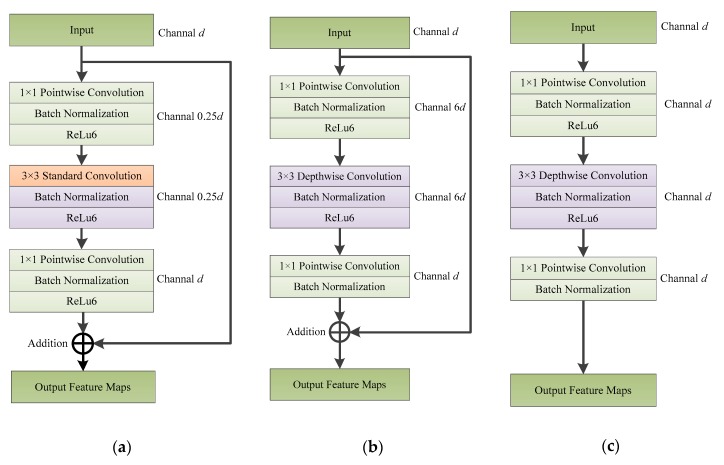
Blocks in a convolutional neural network (CNN): (**a**) standard residual block; (**b**) inverted residual block in MobileNetv2 when stride is 1; (**c**) linear block in MobileNetv2 when stride is 2.

**Figure 7 sensors-20-01999-f007:**
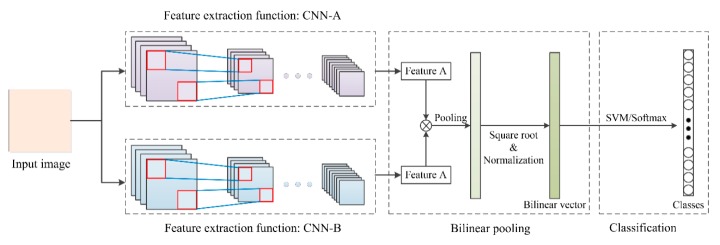
Pipeline of the bilinear-CNN model proposed by Lin et al. [63].

**Figure 8 sensors-20-01999-f008:**
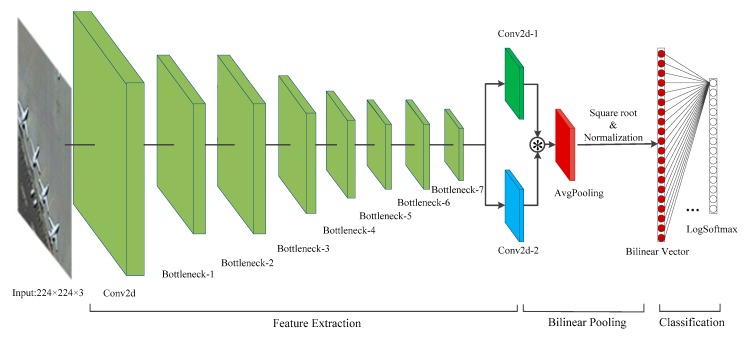
Architecture of BiMobileNet. See text for description.

**Figure 9 sensors-20-01999-f009:**
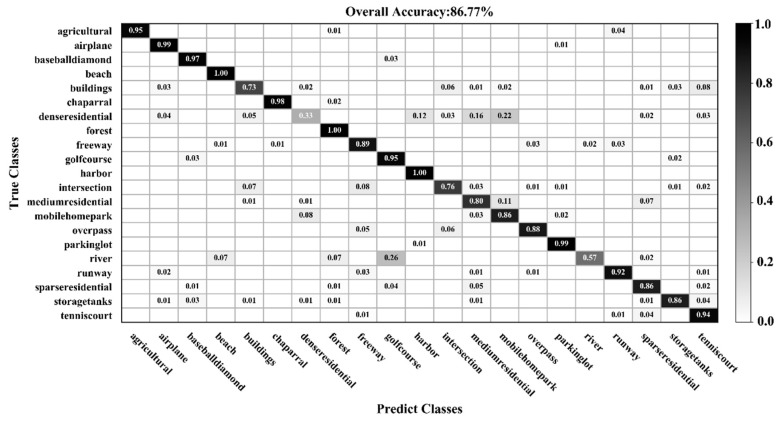
Confusion matrix using a training ratio of 5% on the UC Merced dataset.

**Figure 10 sensors-20-01999-f010:**
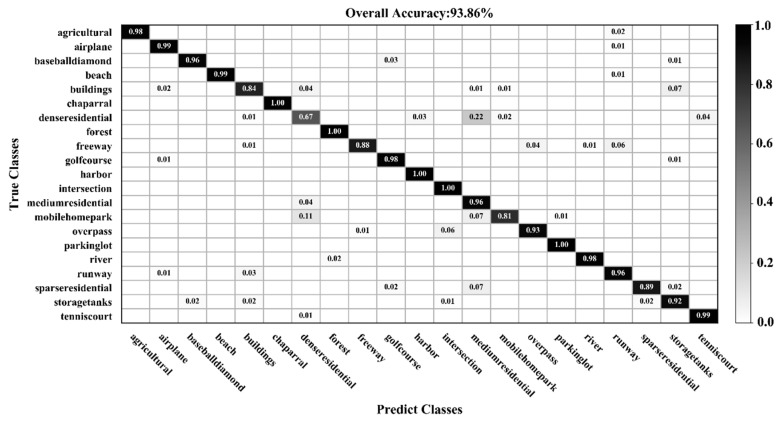
Confusion matrix using a training ratio of 10% on the UC Merced dataset.

**Figure 11 sensors-20-01999-f011:**
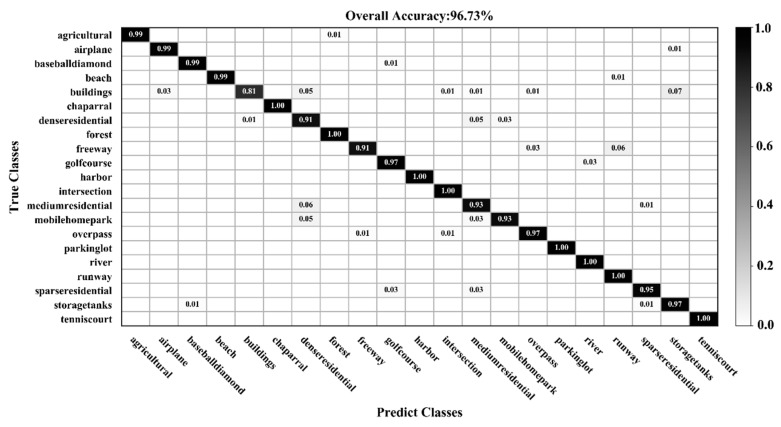
Confusion matrix using a training ratio of 20% on the UC Merced dataset.

**Figure 12 sensors-20-01999-f012:**
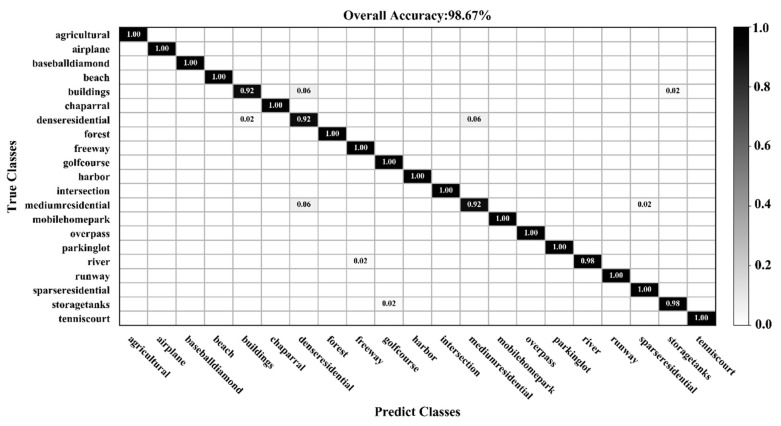
Confusion matrix using a training ratio of 50% on the UC Merced dataset.

**Figure 13 sensors-20-01999-f013:**
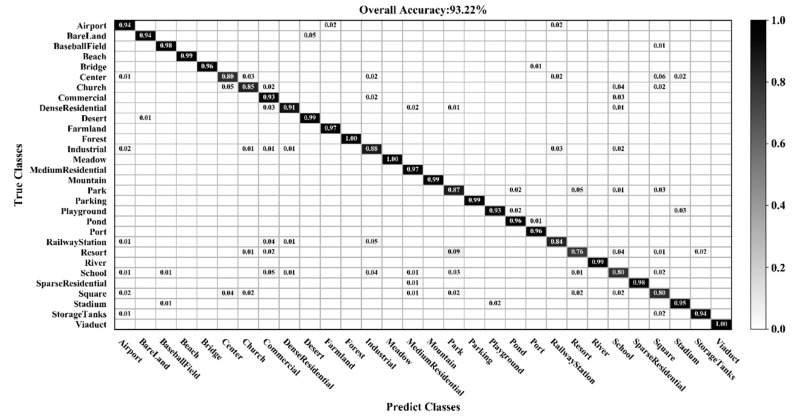
Confusion matrix using a training ratio of 10% on the AID dataset.

**Figure 14 sensors-20-01999-f014:**
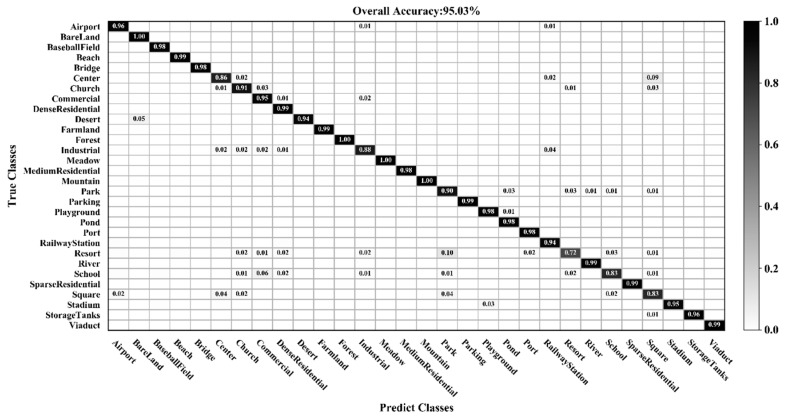
Confusion matrix using a training ratio of 20% on the AID dataset.

**Figure 15 sensors-20-01999-f015:**
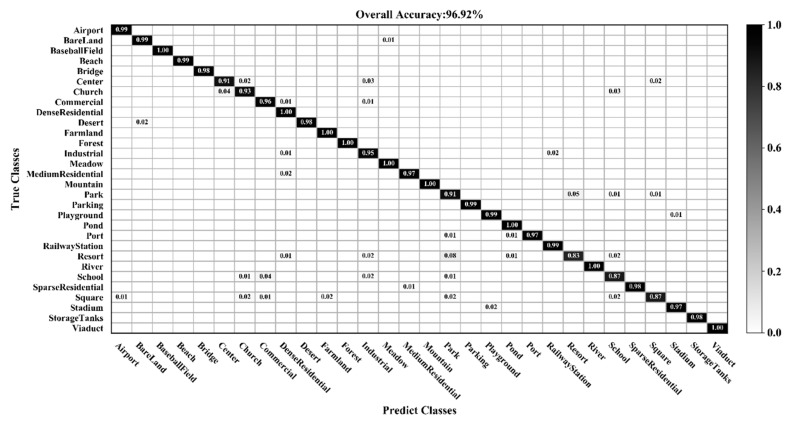
Confusion matrix using a training ratio of 50% on the AID dataset.

**Figure 16 sensors-20-01999-f016:**
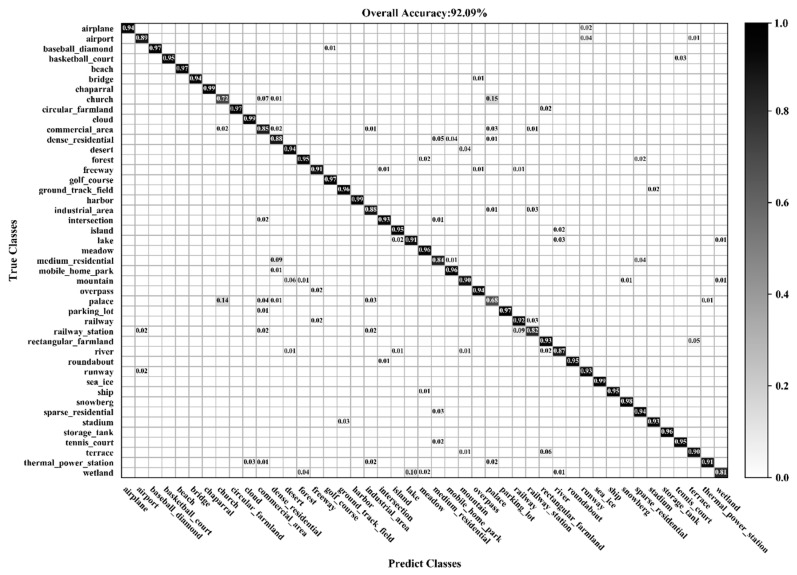
Confusion matrix using a training ratio of 10% on the NWPU-RESISC45 dataset.

**Figure 17 sensors-20-01999-f017:**
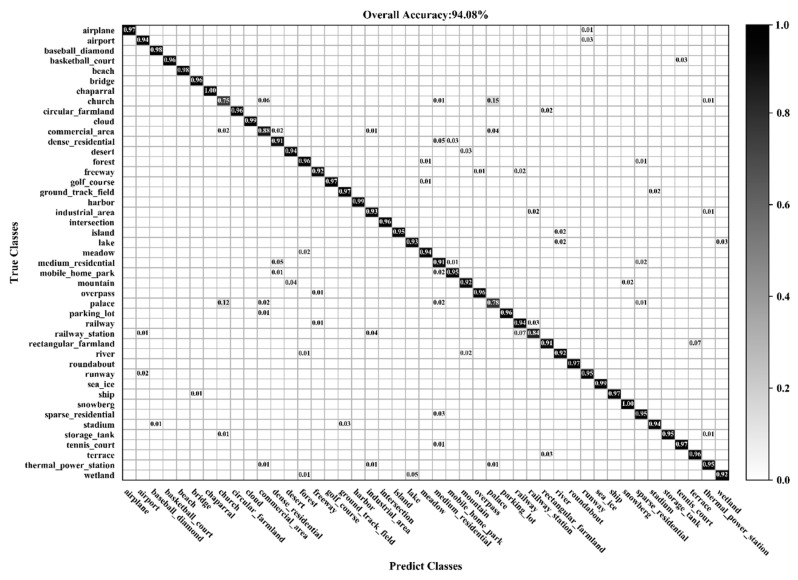
Confusion matrix using a training ratio of 20% on the NWPU-RESISC45 dataset.

**Table 1 sensors-20-01999-t001:** Dataset information.

Dataset Name	Number of Classes	Image Size	Resolution/m	Images per Class	Total Images
UC Merced	21	256 × 256	2	100	2100
AID	30	600 × 600	0.5~8	200~400	10,000
NWPU-RESISC45	45	256 × 256	0.2~30	700	31,500

**Table 2 sensors-20-01999-t002:** BiMobileNet details.

Layer Name	Operation	Input Size	Output Size
Conv2d	Conv2d, kernel size = (3 × 3), stride = 2	224 × 224 × 3	112 × 112 × 32
Bottleneck-1	Linear block, *m* = 1, stride = 1	112 × 112 × 32	112 × 112 × 16
Bottleneck-2	Linear block, *m* = 6, stride = 2	112 × 112 × 16	56 × 56 × 24
	Inverted residual block, *m* = 6, stride = 1	56 × 56 × 24	56 × 56 × 24
Bottleneck-3	Linear block, *m* = 6, stride = 2	56 × 56 × 24	28 × 28 × 32
	Inverted residual block, *m* = 6, stride = 1	28 × 28 × 32	28 × 28 × 32
	Inverted residual block, *m* = 6, stride = 1	28 × 28 × 32	28 × 28 × 32
Bottleneck-4	Linear block, *m* = 6, stride = 1	28 × 28 × 32	28 × 28 × 64
	Inverted residual block, *m* = 6, stride = 1	28 × 28 × 64	28 × 28 × 64
	Inverted residual block, *m* = 6, stride = 1	28 × 28 × 64	28 × 28 × 64
	Inverted residual block, *m* = 6, stride = 1	28 × 28 × 64	28 × 28 × 64
Bottleneck-5	Linear block, *m* = 6, stride = 2	28 × 28 × 64	14 × 14 × 96
	Inverted residual block, *m* = 6, stride = 1	14 × 14 × 96	14 × 14 × 96
	Inverted residual block, *m* = 6, stride = 1	14 × 14 × 96	14 × 14 × 96
Bottleneck-6	Linear block, *m* = 6, stride = 2	14 × 14 × 96	7 × 7 × 160
	Inverted residual block, *m* = 6, stride = 1	7 × 7 × 160	7 × 7 × 160
	Inverted residual block, *m* = 6, stride = 1	7 × 7 × 160	7 × 7 × 160
Bottleneck-7	Linear block, *m* = 6, stride = 1	7 × 7 × 160	7 × 7 × 320
Bilinear Pooling	Conv2d-1, kernel size = (*k* × *k*), stride = 1	7 × 7 × 320	7 × 7 × 1024
	Conv2d-2, kernel size = (*k* × *k*), stride = 1	7 × 7 × 320	7 × 7 × 1024
	AvgPooling kernel size = (7 × 7)	7 × 7 × 1024	1 × 1 × 1024
Classification	Fully Connected	1 × 1 × 1024	class number

**Table 3 sensors-20-01999-t003:** Overall accuracy of the state-of-the-art method on UC Merced dataset. The highest accuracy for each ratio is bolded.

Method	Published Year	Training Ratio		
		20%	50%	80%
BOVW [4]	2010			76.81
VLAT [67]	2014			94.30
MS-CLBP+FV [34]	2016		88.76 ± 0.79	93.00 ± 1.20
TEX-NET-FL (ResNet) [51]	2017		96.91 ± 0.36	97.72 ± 0.54
salM^3^LBP-CLM [54]	2017		91.21 ± 0.75	95.75 ± 0.80
VGG-VD-16 [2]	2017		94.14 ± 0.69	95.21 ± 1.20
CNN-ELM [68]	2017			95.62 ± 0.32
Two-Stream Fusion [69]	2018			98.02 ± 1.03
D-CNN (VGG16) [47]	2018			98.93 ± 0.10
RTN (VGG16) [42]	2018			98.96
DCF (VGG-VD16) [70]	2018		95.42 ± 0.71	97.10 ± 0.85
GCFs+LOFs (VGG16) [9]	2018		97.37 ± 0.44	99.00 ± 0.35
SAL-TS-Net (GoogLeNet) [8]	2018		97.97 ± 0.56	98.90 ± 0.95
Siamese ResNet50 [71]	2019	76.50	90.95	94.29
SF-CNN (VGGNet) [44]	2019			99.05 ± 0.27
VGG16-DF [43]	2019			5298.97
MRBF [72]	2019			94.19 ± 0.15
DDRL-AM (ResNet18) [41]	2019			**99.05 ± 0.08**
WSPM-CRC (ResNet152) [73]	2019			97.95
CTFCNN [52]	2019			98.44 ± 0.58
CapsNet (Inception-v3) [74]	2019		97.59 ± 0.16	99.05 ± 0.24
BiMobileNet (MobileNetv2)	2020	**96.41 ± 0.57**	**98.45 ± 0.27**	99.03 ± 0.28

**Table 4 sensors-20-01999-t004:** Overall accuracy of BiMobileNet under different training ratios. The highest accuracy for each ratio is bolded.

Method	Training Ratio				
	5%	10%	15%	20%	25%
Fine-tuning VGG16	39.53 ± 2.23	53.12 ± 1.15	59.83 ± 2.45	64.68 ± 2.70	69.51 ± 0.65
Fine-tuning ResNet50	39.01 ± 1.62	51.35 ± 1.25	57.40 ± 0.96	64.82 ± 0.64	71.06 ± 0.72
Fine-tuning MobileNetv2	38.64 ± 1.45	52.85 ± 0.85	60.90 ± 1.26	67.86 ± 1.12	72.48 ± 0.40
BiMobileNet	**86.74 ± 1.63**	**93.78 ± 0.75**	**93.90 ± 0.25**	**96.41 ± 0.57**	**97.02 ± 0.55**

**Table 5 sensors-20-01999-t005:** Overall accuracy of the state-of-the-art methods on AID dataset. The highest accuracy for each ratio is bolded.

Method	Published Year	Training Ratio		
		10%	20%	50%
TEX-Net-LF (ResNet) [51]	2017		93.81 ± 0.12	95.73 ± 0.16
salM^3^LBP-CLM [54]	2017		86.92 ± 0.35	89.76 ± 0.45
VGG-VD-16 [2]	2017		86.59 ± 0.29	89.64 ± 0.36
DCA (VGGNet) [57]	2017			91.86 ± 0.28
RTN (VGG16) [42]	2018		92.44	
D-CNN (VGG16) [47]	2018		90.82 ± 0.16	**96.89 ± 0.10**
GCFs+LOFs (VGG16) [9]	2018		92.48 ± 0.38	96.85 ± 0.23
SAL-TS-Net (GoogleNet) [8]	2018		94.09 ± 0.34	95.99 ± 0.35
MRBF [72]	2019			87.26 ± 0.42
SF-CNN (VGGNet) [44]	2019		93.60 ± 0.12	96.66 ± 0.11
CTFCNN [52]	2019			94.91 ± 0.24
WSPM-CRC (ResNet152) [73]	2019			95.11
BiMobileNet (MobileNetv2)	2020	**92.77 ± 0.49**	**94.83 ± 0.24**	96.87 ± 0.23

**Table 6 sensors-20-01999-t006:** Overall accuracy of the state-of-the-art methods on NWPU-RESISC45 dataset. The highest accuracy for each ratio is bolded.

Method	Published Year	Training Ratio	
		10%	20%
Fine-tune VGG16 [3]	2017	87.15 ± 0.45	90.36 ± 0.18
D-CNN (VGG16) [47]	2018	89.22 ± 0.50	91.89 ± 0.22
IOR4 (VGG16) [40]	2018	87.83 ± 0.16	91.30 ± 0.17
RTN (VGG16) [42]	2018	89.90	92.71
DCF (VGG-VD16) [70]	2018	87.14 ± 0.19	89.56 ± 0.25
DDRL-AM (ResNet18) [41]	2018	**92.17 ± 0.08**	92.46 ± 0.09
SAL-TS-Net (GoogLeNet) [8]	2018	85.02 ± 0.20	87.01 ± 0.19
Triplet Network [7]	2018		92.33 ± 0.50
VGG16-DF [43]	2019	89.66	
Siamese ResNet50 [71]	2019		92.28
SF-CNN (VGG16) [44]	2019	89.89 ± 0.16	92.55 ± 0.14
GLANet [46]	2019	91.03 ± 0.18	93.45 ± 0.17
CapsNet (Inception-v3) [74]	2019	89.03 ± 0.21	92.60 ± 0.11
DML (VGG16) [49]	2019	91.73 ± 0.21	93.47 ± 0.30
BiMobileNet (MobileNetv2)	2020	92.06 ± 0.14	**94.08 ± 0.11**

**Table 7 sensors-20-01999-t007:** Accuracy and complexity of state-of-the-art methods on NWPU-RESISC45 dataset. The highest accuracy and least calculation are bolded.

Methods	*λ*	Overall Accuracy		Parameters Numbers (Million)	GFLOPs^1^	Model Size MB
		10%	20%			
Fine-tuning VGG16 [3]	/	87.15 ± 0.45	90.36 ± 0.18	~134.44	~15.60	~512.87
SF-CNN (VGG16) [44]	/	89.89 ± 0.16	92.55 ± 0.14	~17.28	~15.49	~65.93
DML (VGG16) [49]	/	91.73 ± 0.21	93.47 ± 0.30	~134.44	~15.60	~512.87
SAL-TS-Net (GoogLeNet) [8]	/	85.02 ± 0.20	87.01 ± 0.19	~10.07	~1.51	~38.41
BiMobileNet (*k* = 3)	0.50	90.26 ± 0.23	92.77 ± 0.14	3.47	0.17	13.27
	0.75	91.47 ± 0.16	93.64 ± 0.12	5.52	0.33	21.05
	1.00	**92.06 ± 0.14**	**94.08 ± 0.11**	7.76	0.45	29.59
BiMobileNet (*k* = 1)	0.50	90.06 ± 0.11	92.74 ± 0.11	**0.86**	**0.12**	**3.27**
	0.75	91.23 ± 0.09	93.67 ± 0.05	1.59	0.24	6.05
	1.00	91.89 ± 0.19	93.92 ± 0.11	2.52	0.34	9.59

^1^ GFLOPs = 10^9^ floating-point operations; *k* is the kernel size in bilinear pooling layer.

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
