# Peer review of "An Efficient and Lightweight Convolutional Neural Network for Remote Sensing Image Scene Classification"

_sensors, 2020, doi:10.3390/s20071999_

Reviewer 1 Report
In this paper the authors propose BiMobileNet that is the combination of a lightweight network, MobileNetv2, as backbone of a bilinear convolutional model in a remote sensing image classification problem. A good revision of the bibliography is done. The bilinear model is based on the Hadamard product to avoid a very large feature vector. The approach is tested with three datasets and the results are compared with state of the art methods in remote sensing image classification making use of a five times repeated hold-out setup and with different training ratios and always with 100 epochs. A process of data augmentation was carried out to increase the size of the datasets. The proposed approach yields similar results to other more complex methods with a reduced number of parameters to be trained.
The main concerns I have with this paper are the following ones:
- I have misgivings about the proposal in the fact that the use of only one network in the bilinear model and the use of two convolutional layer to feed the bilinear combination, can be refered as bilinear architecture. The strenght of the bilinear architecture is given for the two branches that can produce complementary information. This feature is lost in this proposal because only one network is considered.
- The authors support the proposal into the fact of using a lightweight network because with the fact that with a low number of parameters to train, smaller datasets are necessary to train it and also it can be trained in a mobile device. But in the experiments, the datasets have 2100, 10000 and 31500 images that with data augmentation are increased in size by 6, and even more, they use a pretrained net. So with this number of samples and starting from a pretrained network, more complex architectures can be considered.
- The authors also argue that the use of the lightweight network is because it can be trained in a mobile or edge devices, but the experiments are carried out in a high end GPU as the GTX1080Ti with 11GB of RAM, that is enough to train more complex networks as VGG16 or even RESNET-50.
- In the evaluation protocol, the dataset is divided into train and test and all the networks are trained the same number of epochs which is clearly unfair due to the different complexity of each one. The number of epochs must be different for each one.
- I am agree with the authors that annotating images is a very tedious and time consuming task but if there exists a dataset with annotated images I don't know the reason to not use all of them. So each network in the comparison must be trained with the most amount of images as possible. To avoid overfitting there are many solutions as the introduction of a validation subset.
- With respect to the title of the writing of the paper, many sections can be removed because they are a rewritting of the original paper as sections 2.2.2, 2.2.3 and 2.2.4. Also the extension to explain the Hadamard product is excesive when it is the element-wise matrix product.
- Figure 7 is not original and the source (Lin,2015) is not cited which is a very serious drawback because it could induce to the reader that it has been elaborated by the authors of this paper.
Reviewer 2 Report
Summary:
The paper discussed the use of MobileNetv2, a lightweight convolutional neural network, in combination with bilinear feature manipulation for classifying remote sensing images. Three widely used datasets: UC Merced, AID, and NWPU-RESISC45 were used to evaluate the model.
The literature review properly discussed the relevant areas. The datasets and the method are elaborately discussed as well.
The results successfully show that the model performed better in most cases compared to the other state-of-the-art approaches.
Comments:
The concept of bilinear feature for image classification or MobileNetv2 for lightweight parameters are not novel at all.
Handcraft needs to be replaced by handcrafted.
Author Response
Point 1:The concept of bilinear feature for image classification or MobileNetv2 for lightweight parameters are not novel at all.
Response 1: Bilinear CNN (B-CNN) was first proposed by Lin in his article ‘Bilinear CNN Models for Fine-Grained Visual Recognition’ in 2015 and it was adopted for fine-grained visual recognition such as car recognition and bird recognition. Although the idea of B-CNN is simple, its performance is remarkable. Hoverer, until now, there is few literatures which adopted B-CNN for optical remote sensing scene classification and got significant increasement in accuracy compared to the method in our article. It is the first time that we combined MobileNetv2 and B-CNN in a scene classification task for remote sensing images and got good results. I have made a lot of experiments and find that even the latest convolutional neural networks like EfficientNet can't achieve better performance than MobileNet in a B-CNN. This is an interesting phenomenon. I think it is the unique structure in MobileNet that makes our method so effective. Besides, there are differences and similarities between remote sensing scene classification and fine-grained visual recognition. They both classify images. However, most categories in the fine-grained recognition task are very similar while there are only several categories which are similar in remote sensing scene classification. I think we provide a simple and efficient method with fewer parameters and calculations but better performance for remote sensing scene classification.
Point 2:Handcraft needs to be replaced by handcrafted.
Response 2: Thank you for your suggestion. I made a mistake here. This error has been corrected in revised manuscript.
Reviewer 3 Report
Dear Authors
The paper present a lightweight network for Remote Sensing image classification. The proposed method is really interesting and also this method would be interesting for researchers in the field of study. However there is a paper "A novel spatio-temporal FCN-LSTM network for recognizing various crop types using multi-temporal radar images that presented a lightweight DCNN which I think would be necessary to improve the introduction part by adding this reference.
Author Response
Point 1:The paper present a lightweight network for Remote Sensing image classification. The proposed method is really interesting and also this method would be interesting for researchers in the field of study. However there is a paper "A novel spatio-temporal FCN-LSTM network for recognizing various crop types using multi-temporal radar images that presented a lightweight DCNN which I think would be necessary to improve the introduction part by adding this reference.
Response 1:Thanks for your advice, I ignored those literatures which adopted lightweight DCNN for remote sensing image scene classification because there are only several literatures. In the introduction, the following three literatures are added in the revised manuscript and make the introduction more comprehensive.
Zhang, B.; Zhang, Y.; Wang, S. A Lightweight and Discriminative Model for Remote Sensing Scene Classification with Multidilation Pooling Module. IEEE Journal of Selected Topics in Applied Earth Observations and Remote Sensing, 2019, 12(8):2636-2653.
Zhang, G.; Lei, T.; Cui, Y.; Jiang, P. A Dual-Path and Lightweight Convolutional Neural Network for High-Resolution Aerial Image Segmentation. ISPRS International Journal of Geo-Information, 2019, 8(12): 582.
Teimouri, N.; Dyrmann, M.; Jørgensen, R. A Novel Spatio-Temporal FCN-LSTM Network for Recognizing Various Crop Types Using Multi-Temporal Radar Images. Remote Sensing, 2019, 11(8): 990.
Round 2
Reviewer 1 Report
I want to thank to the authors for effort in improving the paper and while some of the questions and concerns in my previous review comments have addressed, some others are not clearly answered in the authors response.
In response 3 the authors justify the use of a high-end GPU is due to the fact that models are not trained in mobile and edge devices and only they are used for deployment, and I'm totally agree with them. But they don't give a convincing argument in response 5 to use as much as possible to train the network and mix with few-shot and zero-shot learning which is out of scope of this paper. So in my opinion, there is a different argument when decided to use as much as possible from technology with a high-end gpu and on the other hand not to use as much as possible from the samples when they are available.
In response 4, the authors confirm that they don't use a validation set to set the number of epochs and it is based on a stable value of the training loss. Some papers have addressed the fact that using only the training loss or accuracy may induce an overfitting problem.
With respect to response 6, I must admit that I'm completely disagree with authors. I have reread the paper and in sections 2.2.2, 2.2.3 and 2.2.4 there is no mention to remote sensing. They are rewritting of the same sections of the papers where MobileNet a B-CNN are described. So a reference to those papers could be enough. Also in the justification of the explanation of the Hadamard product is not convincing for me. Any reader that can understand the rest of the paper will be able to know which is a element-wise matrix product with no more explanation.
Author Response
Point 1:In response 3 the authors justify the use of a high-end GPU is due to the fact that models are not trained in mobile and edge devices and only they are used for deployment, and I'm totally agree with them. But they don't give a convincing argument in response 5 to use as much as possible to train the network and mix with few-shot and zero-shot learning which is out of scope of this paper. So, in my opinion, there is a different argument when decided to use as much as possible from technology with a high-end gpu and on the other hand not to use as much as possible from the samples when they are available.
Response 1: Thank you for the advice in your comments on the paper. I'm sorry that I didn't make it clear in my last reply. This example of few-shot and zero-shot is given to illustrate that even there are available data and high-end devices for training, not using most of the data for training is still meaningful. Of cause, in the field of deep learning, the more data, the better. But when faced with a new dataset, if we can only label a small portion of the data for training and then predict the remaining data, which can save a lot of labour and time costs. That's our purpose. In addition, a model which needs not so much training data has better performance is a better choice no matter there is a high-end gpu or not.
Point 2: In response 4, the authors confirm that they don't use a validation set to set the number of epochs and it is based on a stable value of the training loss. Some papers have addressed the fact that using only the training loss or accuracy may induce an overfitting problem.
Response 2: First, the division of the dataset (training set for training and test set for validation) is set according to other state-of-the-art methods and the evaluation of the results is objective.Second, although we use the training loss or accuracy to decide whether to continue training, we use the test set to verify that the model is overfitting or not. The test set is not involved in the training of the model and there is no overlap between the test set and the training set. Only the training loss or accuracy may induce an overfitting problem, but when compared to other methods, the accuracies of the article are obtained on the test set. When the training loss or accuracy is stable for dozens of epochs and then we save the weights of model at last ten epochs to validate the test set and get the average accuracy. We repeated this experiment five times by randomly dividing the dataset into training set and test set five times. The overall accuracy is obtained by averaging the five results. The overall accuracy is superior to most state-of-the-art methods, and I don't think our method is obviously over-fitting.
Point 3: With respect to response 6, I must admit that I'm completely disagree with authors. I have reread the paper and in sections 2.2.2, 2.2.3 and 2.2.4 there is no mention to remote sensing. They are rewriting of the same sections of the papers where MobileNet and B-CNN are described. So, a reference to those papers could be enough. Also, in the justification of the explanation of the Hadamard product is not convincing for me. Any reader that can understand the rest of the paper will be able to know which is an element-wise matrix product with no more explanation.
Response 3: As I responded last time, it is the first time that we combined MobileNetv2 and B-CNN for remote sensing image scene classification and achieved significant improvement in accuracy. We gave the details of BiMobileNet in section 2.3 which adopted some modules from B-CNN and MobileNet. I think it is necessary to briefly introduce relevant knowledge in the article, especially for those in the field of remote sensing who don't know B-CNN or MobileNet. Hadamard product maybe excessive and the redundancy has been deleted in section 2.3, Line 292.